# Revisiting the Dissolution of Praziquantel in Biorelevant Media and the Impact of Digestion of Milk on Drug Dissolution

**DOI:** 10.3390/pharmaceutics14102228

**Published:** 2022-10-19

**Authors:** Thomas Eason, Gisela Ramirez, Andrew J. Clulow, Malinda Salim, Ben J. Boyd

**Affiliations:** 1Drug Delivery Disposition and Dynamics, Monash Institute of Pharmaceutical Sciences, Monash University (Parkville Campus), 381 Royal Parade, Parkville, VIC 3052, Australia; 2BioSAXS Beamline, Australian Synchrotron, ANSTO, 800 Blackburn Road, Clayton, VIC 3168, Australia; 3Department of Pharmacy, University of Copenhagen, Universitetsparken 2, 2100 Copenhagen, Denmark

**Keywords:** praziquantel, Biltricide, dissolution, biorelevant media, digestion, milk

## Abstract

Praziquantel is a poorly water-soluble drug used to treat parasitic infections. Previous studies have suggested that its rate and extent of dissolution in milk and biorelevant media are slow and limited compared to dissolution in the pharmacopoeial-recommended medium, despite being reported as displaying a positive food effect upon administration. This study aimed to revisit the dissolution of praziquantel in biorelevant media and milk to better understand this apparent dichotomy. The context of digestion was introduced to better understand drug solubilisation under more relevant gastrointestinal conditions. The amount of praziquantel solubilised in the various media during digestion was quantified using high performance liquid chromatography (HPLC) and the kinetics of dissolution were confirmed by tracking the disappearance of solid crystalline drug using in situ small angle X-ray scattering (SAXS). For the dissolution media, where sodium lauryl sulfate (SLS) is typically included as a wetting agent, a prominent effect of SLS on drug dissolution was also apparent where >2.5 fold more drug was solubilised in SLS-containing dissolution medium compared to that without (0.1 M HCl only). In milk, significant dissolution of praziquantel was observed only during digestion and not during dispersion, hence suggesting that (1) milk can be potentially administered with praziquantel to improve oral bioavailability and (2) incorporating a digestion step into existing in vitro dissolution testing can better reflect the potential for a positive food effect when lipids are present.

## 1. Introduction

Praziquantel (2-(cyclohexylcarbonyl)-1,2,3,6,7,11b-hexahydro-4H-pyrazino [2,1-a]isoquinolin-4-one; Figure 1) is an anti-parasitic drug in the antihelmintics drug class. It is used to treat intestinal and urogenital schistosomiasis, also known as snail fever: a parasitic disease caused by blood flukes whose larva can penetrate skin when in contact with infested water [1]. This tropical disease is prevalent in developing countries, where up to 60% of young children are infected [2]. The use of praziquantel in children has been shown to be safe but typically requires crushing of tablets and administration of the powders as suspensions with various fluids [2]. Understanding how fluid and food intake affect oral bioavailability of praziquantel in children is therefore of paramount importance.

Being classified in the Biopharmaceutics Classification System (BCS) as a class 2 drug [3], praziquantel is a poorly water-soluble drug that has insufficient dissolution and solubility in the gastrointestinal tract, which limits its oral absorption [4]. Re-formulating praziquantel as various dosage forms including solid lipid nanoparticles [5,6] and solid dispersions [7] are amongst some of the efforts explored in an attempt to improve oral bioavailability of praziquantel. Although a high fat or high carbohydrate meal has been shown to improve the oral bioavailability of praziquantel in adult volunteers, translating the findings to infants can be challenging given the different gastrointestinal physiology. Praziquantel has not been approved for children under the age of four and it is not clear whether the market for older children would be sufficient to warrant development of an oral suspension formulation of praziquantel. The positive effect of food on bioavailability was also not reflected in in vitro dissolution studies which also complicates translatability. In particular, the in vitro dissolution of praziquantel in milk, as a potential paediatric-friendly vehicle, was found to be slower than in the simulated fasted and fed small intestinal fluids [8].

The essence of the current study is summarised schematically in Figure 1. The dissolution of praziquantel was revisited in different biorelevant media including pharmacopoeial-recommended fluid and milk to (1) compare the dissolution behavior of praziquantel in biorelevant and USP-recommended media; and (2) to better understand the potential for milk to be used as a vehicle for praziquantel to enable a paediatric-friendly lipid formulation, potentially reduce the current dosage burden and improve oral bioavailability. Milk is a staple diet for children that typically contains 3–4% fat of which 98% are triglycerides, the digestion of which by gastrointestinal lipases produces monoglycerides and fatty acids which has previously been shown to enhance the solubilisation of a range of BCS class 2 compounds [9]. High performance liquid chromatography (HPLC) was used to quantify the amount of praziquantel dissolved and time resolved small angle X-ray scattering (SAXS) was used to probe the rate of solubilisation of crystalline drug into media.

## 2. Materials and Methods

### 2.1. Materials

Biltricide^®^ tablets (trade name for praziquantel), supplied by Alta Biosciences and manufactured by Bayer AG, Leverkusen, Germany, were provided by the Bill and Melinda Gates Foundation. Each tablet contains 600 mg of praziquantel and the following excipients: corn starch, magnesium stearate, microcrystalline cellulose, povidone, sodium lauryl sulfate (SLS), polyethylene glycol, titanium dioxide and hypromellose [1]. Praziquantel active pharmaceutical ingredient (API; Pharmaceutical Secondary Standard, PHR1391), 4-bromophenylboronic acid (4-BPBA, >95%), and 1-methyl-2-pyrrolidinone (biotech. Grade, ≥99.7%) were purchased from Sigma Aldrich (St. Louis, MO, USA). Bovine milk (Pura brand, 3.4% fat or Pauls brand, 3.8% fat) was purchased from a local supermarket (Melbourne, VIC, Australia). Infant formula was provided by Medicines for Malaria Venture (Geneva, Switzerland). Nutritional information of the milk and infant formula was tabulated in Appendix A [10]. FaSSIF/FeSSIF/FaSSGF powder (for preparation of Fasted Simulated Small Intestinal Fluid, Fed Simulated Small Intestinal Fluid and Fasted Simulated Gastric Fluid, respectively) was purchased from Biorelevant.com Ltd. (London, UK). Pancreatin (USP grade from porcine pancreas) was purchased from Southern Biological (Alphington, VIC, Australia). Sodium hydrogen phosphate (Na_2_HPO_4_) was purchased from Chem-Supply (Gillman, SA, Australia). Sodium lauryl sulfate (96%) was purchased from Huntsman Corporation (West Footscray, VIC, Australia). Hydrochloric acid (36%) was purchased from LabServ (Longford, Ireland). Acetonitrile (chromatography grade) was purchased from Merck (Darmstadt, Germany). Water was sourced from a MilliQ^®^ water system (Millipore, Sydney, NSW, Australia).

### 2.2. Dissolution of Praziquantel in USP-Medium and Biorelevant Media

In vitro dissolution of Biltricide^®^ was performed using a USP dissolution (paddle) apparatus 2. Biltricide^®^ tablet (1 tablet) was dispersed in 900 mL of USP-recommended dissolution medium (0.1 M HCl in water with 2 mg/mL SLS) [11]; 0.1 M HCl in water with no added SLS; 50 mM phosphate buffer pH 6.8; and fasted/fed simulated small intestinal fluids (FaSSIF/FeSSIF) in phosphate buffer at 37 °C. Paddle speed was 50 rpm. Phosphate buffer was prepared by dissolving the appropriate amount of Na_2_HPO_4_ in water and the pH was adjusted to 6.8. FaSSIF and FeSSIF solutions were prepared in phosphate buffer using FaSSIF/FeSSIF/FaSSGF powder where the final concentration of bile salt (taurocholate): phospholipids in FaSSIF and FeSSIF was 3:0.75 mM and 15:3.75 mM, respectively. During dissolution, 1 mL of sample was collected at each time point up to 1 h and filtered through a 0.45 µm syringe filter (13 mm Durapore^®^ PVDF low protein binding, Merck). A 0 min time point was also collected prior to the addition of Biltricide^®^ tablet. Filtered samples were stored at −20 °C until further analysis.

Dissolution of Biltricide^®^ in the gastric medium followed by a pH switch to small intestinal pH at 6.8 was also performed. Biltricide^®^ tablet (1 tablet) was dispersed in the gastric medium (either 800 mL of 0.1 M HCl in water with 2 mg/mL SLS or 900 mL of FeSSIF pH 1.2) for 1 h after which the pH was adjusted to 6.8 using 5 M NaOH. Following pH adjustment, 80 mL of 10 × concentrated FaSSIF was added to the former (0.1 M HCl with 2 mg/mL SLS). Samples were collected and filtered as described above prior to further analysis.

### 2.3. Dissolution of Praziquantel in Milk, Infant Formula, and the Impact of Digestion

Pancreatic lipase solution used for digestion was prepared by adding 25 mL phosphate buffer, pH 6.8 to 20 g pancreatin powder followed by centrifugation at 4444× *g* for 15 min at 4 °C. The collected supernatant was used as is. For dissolution of praziquantel in milk or infant formula, 1 Biltricide^®^ tablet was added to 800 mL of milk or reconstituted infant formula (both containing 3.4% fat) at 37 °C. After 1 h of dispersion, 90 mL of pancreatic lipase solution was added to initiate the digestion of milk or infant formula. Paddle speed was set to 200 rpm to enhance mixing of the milk dispersion, which has a higher viscosity relative to buffer. The pH of the digesting formulations was maintained at 6.8 using 2 M NaOH titrant on an automated dosing unit controlled using a pH-STAT controller (Metrohm 902 STAT titrator). Samples (200 µL) were collected at specified time points during the initial 1 h dispersion phase and during the subsequent 1 h digestion phase. For the milk or infant formula samples collected during digestion, 2 µL of 4-BPBA (0.5 M in methanol) was added to inhibit digestion. Ultracentrifugation of the samples was then performed at 243,277× *g* for 5 min at 37 °C on an Optima MAX-TL ultracentrifuge, TLA-110 rotor (Beckman Coulter, Indianapolis, IN, USA). Following ultracentrifugation, the upper lipid phase and the aqueous supernatant phase were collected, combined, and stored at −20 °C until further analysis.

### 2.4. Solubility of Praziquantel in USP-Medium, Milk, Infant Formula and Biorelevant Media

Biltricide^®^ (1 tablet) was added to 20 mL of the dissolution media (0.1 M HCl in water with and without 2 mg/mL SLS, phosphate buffer pH 6.8, FaSSIF, FeSSIF, milk, infant formula, pre-digested drug-free milk, or pre-digested drug-free infant formula) and incubated at 37 °C with constant mixing. Samples were collected at specified time points from 4 to 48 h and were either filtered using a 0.45 µm syringe filter (for non-milk media) or centrifuged (for milk) as described in Section 2.2 and Section 2.3.

### 2.5. Quantification of Praziquantel Using HPLC

Praziquantel collected in filtrate (in non-milk media) and supernatant after ultracentrifugation (in milk) was quantified using HPLC. Briefly, samples collected after filtration were diluted 20× with the HPLC mobile phase (60% acetonitrile in water) before being subjected to chromatographic separation. The aqueous and lipid fractions of the collected milk (and digested milk) samples were diluted with the HPLC mobile phase and centrifuged at 16,162× *g* for 30 min. The supernatant was collected and analysed using HPLC. Chromatographic separation of praziquantel was performed on a Waters Symmetry C18 column (4.6 mm ID, 75 mm length, 3.5 µm particle size and 100 Å pore size) at 40 °C with isocratic elution in the aforementioned mobile phase and detected using a UV detector at 220 nm (SPD-20A UV, Shimadzu Corporation, Kyoto, Japan). Run time was 6 min and retention time of praziquantel was about 2.9 min. Injection volume was 2 µL. The HPLC system consisted of a Shimadzu CBM-20A controller system, an LC-20AD solvent delivery module, an SIL-20A auto-sampler and a CTO-20A column oven. Praziquantel standards were prepared using praziquantel API, where a stock solution (1 mg/mL in 1-methyl-2-pyrrolidinone) was diluted with the mobile phase to a final concentration range of 0.1–40 µg/mL.

### 2.6. Dissolution of Praziquantel as Probed Using Small Angle X-ray Scattering (SAXS)

The SAXS/WAXS beamline at the Australian Synchrotron (ANSTO, Clayton, VIC, Australia) [12] was used to monitor the progression of drug solubilisation during digestion of milk. Crushed Biltricide^®^ tablet was added to 18 mL of milk or infant formula in specific ratios of equivalent praziquantel API to fat to enable the determination of the impact of fat content on drug solubilisation during digestion. The mixtures of drug and milk/reconstituted infant formula were transferred into a digestion vessel maintained at 37 °C by a circulating water bath. Digestion was initiated by remote injection of 2 mL pancreatic lipase solution prepared in phosphate buffer, pH 6.8. Activity of the lipase was about 700 tributyrin unit/mL of digest. The pH of the digesting medium was controlled and maintained at 6.5 using a pH-STAT controller (Metrohm 902 STAT titrator). The digesting sample was aspirated from the digestion vessel through a peristaltic pump to a free-standing quartz capillary (external diameter 1.5 mm) mounted in the X-ray beam (15 keV, wavelength = 0.827 Å). The sample to detector distance was approximately 1220 mm to cover the *q* range between 0.02 and 1.85 Å^−1^. 2D SAXS patterns were recorded using a Pilatus 2M detector with 5 s acquisition time and 15 s delay between each measurement. The raw data were reduced to I(*q*) vs. *q* by radial integration using ScatterBrain version 2.71. *q* is the length of the scattering vector defined by *q* = (4π/λ)sin(2θ/2) where λ is the X-ray wavelength and 2θ is the scattering angle. The X-ray scattering pattern of Biltricide^®^ powder was also measured in a glass capillary mounted in the X-ray beam. The area under a characteristic Bragg peak for praziquantel with strong intensity was integrated and plotted as a function of time using Origin 2017 software (OriginLab Corporation, Northampton, MA, USA).

## 3. Results and Discussion

### 3.1. Dissolution of Praziquantel in Non-Milk Media

Orally administered drugs are subjected to a range of physiological conditions through the gastrointestinal tract that depend on age, gastrointestinal function, and food intake. Assessing the dissolution of drugs in simulated gastrointestinal conditions in vitro is therefore imperative as a first step to predict the performance of the drug formulations in vivo. Dissolution methods for a range of drugs recommended by the U.S. Pharmacopeia (USP) and U.S. Food and Drug Administration (FDA) can be found in the Dissolutions Methods Database [11,13]. The conditions (pH, volume and surfactant additives in the dissolution medium, type of USP apparatus and the rotation speed) are drug and dosage-form dependent. The recommended dissolution medium for praziquantel tablets is 0.1 M HCl with 2 mg/mL SLS added as a wetting agent. A previous study showed that for a 600 mg dose tablet, >90% praziquantel was released after 1 h dissolution in the recommended pharmacopoeial-dissolution medium, while less than 60% drug was released in small intestinal conditions depending on the concentrations of bile salts [8]. It was therefore unclear whether the presence of SLS can serve as a surrogate for bile salts to predict drug solubilisation.

In this study, we confirm that addition of SLS to the dissolution media (0.1 M HCl) increased the amount of praziquantel solubilised by greater than 2.5 fold compared to 0.1 M HCl with no SLS due to the micellar solubilisation capacity of the surfactant (Figure 2a). Praziquantel is a poorly water-soluble neutral compound [3], thus the solubility in gastric (0.1 M HCl) and small intestinal (50 mM phosphate buffer, pH 6.8 with no bile salts) conditions were not statistically different (297 ± 6 µg/mL and 282 ± 149 µg/mL, respectively, Table 1). Our findings appear slightly higher but probably not significantly different to those reported in the literature where the solubility of praziquantel in 0.1 M HCl and 200 mM phosphate buffer pH 6.8 was 164 µg/mL and 206 µg/mL, respectively [14], which could be due to differences in the sample incubation temperatures (23 °C vs. 37 °C in the current study) and the presence of surfactant excipients in the Biltricide^®^ tablets used in this study. In addition, as depicted in Figure 2b, once praziquantel was dissolved in the low pH gastric condition, no drug precipitation occurred when the pH of the medium was adjusted to 6.8, suggesting that praziquantel could remain solubilised during gastrointestinal transit.

To evaluate the effects of bile on drug dissolution and solubility, biorelevant media were used at 3 mM bile salt/0.75 mM phospholipids (FaSSIF) and 15 mM bile salt/3.75 mM phospholipids (FeSSIF) to simulate fasted and fed conditions, respectively. Figure 2a shows the apparent effect of bile salts and phospholipids on drug dissolution where a greater amount of praziquantel was solubilised in FeSSIF compared to FaSSIF and phosphate buffer; whilst no significant differences were observed between FaSSIF and phosphate buffer. Similar behavior was observed in the solubility measurements of praziquantel in the biorelevant media (Table 1). These findings suggest that secretion of bile salts at fed state concentrations may increase the apparent solubility of praziquantel and that micelles in FeSSIF [15] can provide a greater capacity for drug to be solubilised compared to vesicles in FaSSIF [15]. However, in neonates and infants the concentration of bile salts in the fed state is much lower at approximately 1 mM [16], indicating that fed state conditions with respect to bile concentrations alone may not be sufficient to enable praziquantel to remain completely solubilised under intestinal conditions.

### 3.2. Dissolution of Praziquantel in Milk-Based Media

The results in Section 3.1 indicate that neither changing pH nor adding bile salt can stimulate dissolution beyond approximately 50%. Conceptually this is contrasted with the known positive food effect for praziquantel. This highlights the problem that biorelevant media may not provide sufficient solubilisation capacity to completely reflect the behaviour expected of a drug that shows a positive food effect, because in the case of taking a meal, the digested lipid provides additional solubilisation capacity over the bile salt micelles. Therefore, improving the oral bioavailability of praziquantel in infants is unlikely to be achieved only through addition of bile as a fed state response, but would require appropriate formulations and/or the presence of food. Given the poorly water-soluble nature of praziquantel, it is not surprising that administration of praziquantel with a high fat or high carbohydrate meal increases the oral bioavailability, as has been shown in several in vivo studies on healthy adult volunteers [4,17]. However, as infants are typically fed on liquid and/or soft food, it is important to understand how these food vehicles can be used to improve oral bioavailability of the co-administered drug [18].

Previous studies have shown that milk can be used as a potential formulation to improve the solubilisation of several poorly water-soluble drugs that can lead to greater drug exposure and oral bioavailability [9,19,20]. Effects of co-administering praziquantel with milk on drug dissolution are however not understood and not widely explored [8,21]. Dinora et al. showed slow dissolution of praziquantel in 2.8% fat milk where only 16% drug (from 600 mg dose) was dissolved in 500/900 mL milk after 1 h, compared to about 27% and 55% in FaSSIF and FeSSIF, respectively [8]. This finding in isolation, without considering digestion, would indicate that milk may not be a useful vehicle for the drug. In separate studies, Trastullo et al. compared the percentage of praziquantel dissolved (formulated as granules) in 3.6% fat milk and water and found that about 14% (from 320 mg dose) and 11% of the praziquantel was dissolved in 200 mL of the respective media after 30 min [21]. These observations collectively would lead to the prediction of a lack of food effect on drug dissolution, which is inconsistent with studies in human volunteers and we hypothesise that this is due to the studies ignoring the impact of digestion on drug solubilisation when administered in milk [9].

To clarify the effects of digestion of milk on the dissolution of praziquantel, pancreatic lipase was added to the milk (3.4% fat) in the USP apparatus to initiate the process of digestion. Figure 3a shows that the percentage of praziquantel dissolved was significantly higher when digestion was initiated at 60 min compared to dispersion without digestion. Dispersion could only result in approximately 40% of the drug being dissolved after 1 h, but the drug was completely solubilised after just 15 min of digestion. Comparatively in the case of infant formula, a greater amount of drug was solubilised during dispersion where approximately 70% of the dose was dissolved after 1 h (Figure 3b) and a complete drug solubilisation was achieved after 15 min of digestion. These findings are not unexpected given the higher drug solubilisation capacity in infant formula compared to milk as reflected from the solubility measurements summarised in Table 1. The association between drug solubilisation and the types of milk-based media used has also been reported for a range of other poorly water-soluble drugs [10,22,23], consistently showing an influence of the fatty acid and monoglyceride chain length distribution on the amount of drug solubilised during digestion. Thus, the mechanism proposed here is that production of fatty acids and monoglycerides as a result of lipid digestion provides a medium with much greater solubilisation capacity (dispersion + digestion profile in Figure 2a), driving dissolution of drug that otherwise would remain undissolved (e.g., dispersion only data in Figure 2a).

The effect of digesting milk fat on the solubilisation of praziquantel was also determined using small angle X-ray scattering (SAXS). SAXS has been used recently to follow the solubilisation of drugs during digestion of lipid formulations (including milk and infant formula) by tracking changes in the amount of crystalline drug in suspension during digestion. A decrease in the intensity of Bragg peaks characteristic of the crystalline drug during digestion (not dispersion) indicates solubilisation is occurring as a consequence of digestion. The scattering is able to be acquired with a time resolution on the order of seconds to minutes, i.e., physiologically relevant time scale, by using a synchrotron X-ray source [9,10,19,24]. It is worth noting that these digestion experiments were not performed in a USP dissolution vessel, but scaled down to a smaller volume in vitro digestion apparatus. The characteristic X-ray scattering pattern of praziquantel is shown in Figure 4a. The area under the Bragg peak at *q* = 1.20 Å^−1^ was used to quantify crystalline drug; the strongest intensity peak at *q* = 0.28 Å^−1^ was not used due to interference from Bragg peaks associated with lamellar phases that form during the digestion of milk [25]. Analysis of the characteristic Bragg peak at *q* = 1.20 Å^−1^ during digestion of praziquantel in milk (Figure 4b) showed a strong decrease in peak area compared to in the absence of digestion. Similar trend in drug solubilisation was observed across different peak positions, an example of which is shown in Appendix A. Crystalline drug remained in suspension at the latter stages of digestion with 87.7 mg drug/g milk fat (60 mg API equivalent dose in 18 mL of 3.8% fat milk) indicating incomplete drug solubilisation; but complete drug solubilisation was achieved with 58.5 mg drug/g milk fat (Figure 4b). These results suggest that the range between which praziquantel could be solubilised into the digested milk was between 58.5 and 87.7 mg drug/g milk fat.

In separate studies, the effectiveness of infant formula for solubilising praziquantel during digestion was also examined. The same infant formula, which contains mixtures of medium chain [10] and long-chain triglycerides has been shown to enhance the solubilisation of anti-malarial drugs [10,22] and a riminophenazine antibiotic [23]. Figure 4b shows that the degree of praziquantel solubilisation in the infant formula is different to milk, with the infant formula exhibiting a higher level of drug solubilisation potentially due to differences in lipid composition [27]. Near complete solubilisation was achieved at drug/fat ratio of 133 mg/g, which is equivalent to the consumption of about 20 mL of reconstituted 3.8% fat infant formula and a 100 mg dose of praziquantel by a 5 kg weight infant (assuming a prescribed dosage of 20 mg dose/kg for schistosomiasis [1]). Of note is the higher capacity for dissolution of drug in the dynamic digestion format compared to the equilibrium solubility, which may be attributed to continual changes in lipid composition during the digestion process where transient mixed lipid compositions may provide a solubilising environment of greater capacity than the final fully digested formulation. Therefore, given that praziquantel is considered a BCS class 2 drug and shows a positive food effect [4,17,28] fat-containing excipients should increase bioavailability. In the context of pediatrics, infant formula could therefore be considered as a viable alternative formulation to milk for administration of praziquantel to children, in order to improve the oral bioavailability in neonates and infants. This could take a pediatric friendly format, such as a powder blend in a sachet or a dispersible tablet containing praziquantel and infant formula. Although the regulatory hurdles around the use of infant formula remain [29], the increasing interest in these materials as delivery vehicles when digestion is included in the drug solubilisation assessment provides additional important tools for the formulation of poorly water soluble drugs especially for low economy and paediatric settings.

## 4. Conclusions

Intake of a high fat or high carbohydrate meal has been shown previously to enhance the oral bioavailability of praziquantel but this behavior was not reflected in the in vitro dissolution testing of milk. In this study, we show that while praziquantel can be solubilised to a limited extent into bile salt/phospholipid micelles, the drug exhibits higher solubilisation capacity into the digestion products of milk and infant formula. Incorporating a digestion step into in vitro dissolution testing, particularly for digestible lipid-based formulations is therefore anticipated to improve the link between in vitro dissolution/digestion and in vivo exposure. Future in vivo studies are needed to confirm the relationship. Our studies also show that inclusion of sodium lauryl sulfate in the dissolution media (a surfactant in the recommended USP-dissolution medium for praziquantel) does not reflect the behavior of bile salt micelles in in vitro dissolution tests by dramatically enhancing drug solubility compared to the bile salt media, which will be exaggerated when simulating gastrointestinal conditions for young children where the concentration of bile salt is lower than in adults.

## Figures and Tables

**Figure 1 pharmaceutics-14-02228-f001:**
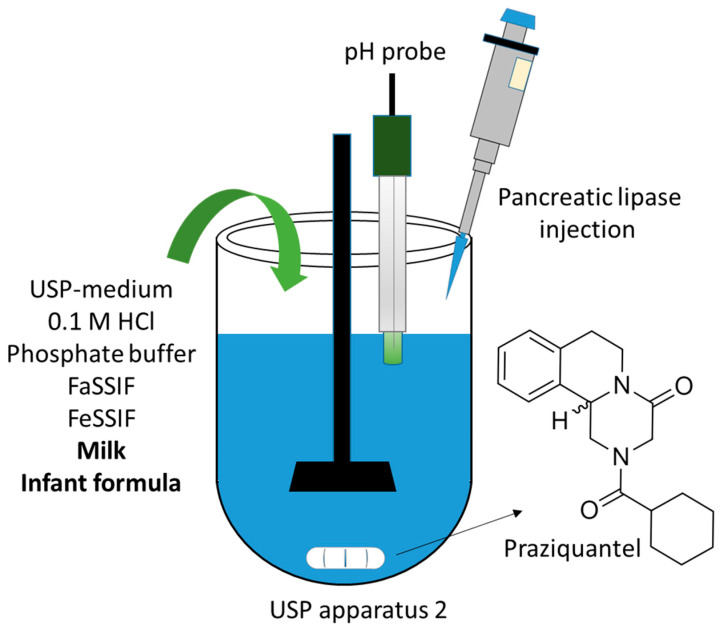
Schematic representation of the concept of this study. Dissolution of praziquantel was studied in a range of different media including in milk and infant formula during digestion.

**Figure 2 pharmaceutics-14-02228-f002:**
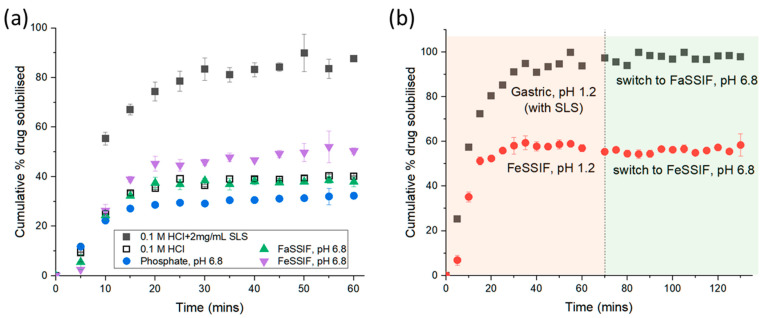
(**a**) Percentage of praziquantel dissolved during in vitro dissolution testing of 600 mg praziquantel Biltricide^®^ tablet in 900 mL of different dissolution media. Errors are standard deviation from n = 3 replicates. (**b**) Effects of pH switching on percent of drug solubilised.

**Figure 3 pharmaceutics-14-02228-f003:**
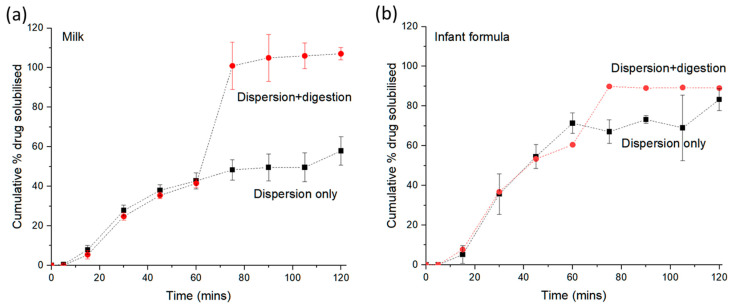
Solubilisation of praziquantel during either dispersion alone or dispersion followed by digestion in (**a**) milk and (**b**) infant formula. Biltricide^®^ (600 mg praziquantel tablet) was dispersed in 800 mL of milk or infant formula (black square; ‘dispersion only’, n = 3) and for the digestion data set, pancreatic lipase was added after 60 min dispersion to initiate digestion (red circle; ‘dispersion + digestion’, n = 2 for infant formula). Dashed lines are drawn as a guide to the eye.

**Figure 4 pharmaceutics-14-02228-f004:**
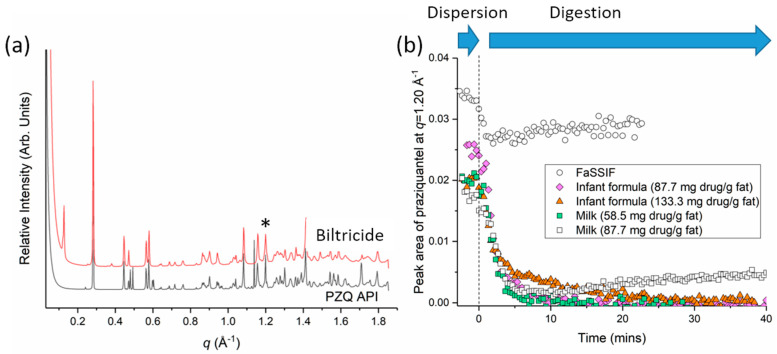
(**a**) X-ray scattering patterns of crushed Biltricide^®^ tablet and praziquantel API powder. Additional Bragg peaks in the API powder at positions *q* = 0.480, 0.493, 0.598, 0.603, and 1.140 Å^−1^ suggest that it is a non-racemic compound [26]. (**b**) Changes in peak area during digestion for the diffraction peak at *q* = 1.20 Å^−1^ (* in panel (**a**)) in FaSSIF, milk and infant formula at different drug to fat ratios (n = 1). Pancreatic lipase was injected at 0 min to initiate digestion.

**Table 1 pharmaceutics-14-02228-t001:** Solubility of praziquantel in different dissolution media. Errors are standard deviation from n = 3 replicates.

Dissolution Media	Solubility of Praziquantel ± SD (µg/mL)	Solubility of Praziquantel ± SD (mg/g Fat)
0.1 M HCl + 2 mg/mL SLS (USP)	546 ± 138	-
0.1 M HCl	297 ± 6	-
Phosphate buffer	282 ± 149	-
FaSSIF	289 ± 77	-
FeSSIF	581 ± 20	-
Milk (3.4% fat)	372 ± 15	10.9 ± 0.4
Digested milk	1782 ± 1131	52.4 ± 33.3
Infant formula (3.4% fat)	674 ± 11	19.8 ± 0.3
Digested infant formula	1744 ± 781	51.3 ± 23.0

## Data Availability

The data presented in this study are available in this article (and Appendix A).

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
