# Peer review of "Revisiting the Dissolution of Praziquantel in Biorelevant Media and the Impact of Digestion of Milk on Drug Dissolution"

_pharmaceutics, 2022, doi:10.3390/pharmaceutics14102228_

Round 1

Reviewer 1 Report (Previous Reviewer 5)

1. Fig 2 and 3, on y axis it should be percent drug release.

2. Justify sudden increase in drug release after 60 minutes in Figure 3.

3. Figure 4 to be discussed and correlated properly. Cite and compare with literature reports.

4.  The outcome of the study to be corelated with therapeutic outcome.

Author Response

Reviewer 1

1.1. Fig 2 and 3, on y axis it should be percent drug release.

Response: It is not a release process, it is dissolution/solubilisation. We use the term solubilisation so that is consistent with the situation in Figure 3, because the drug is highly lipophilic so it is mostly dissolved into the lipid droplets in milk. Consequently, no change was made in response to this comment.

1.2. Justify sudden increase in drug release after 60 minutes in Figure 3.

Response: The reason is completely described in the text (digestion of the milk results in a boost in solubilising capacity of the medium), however to clarify the word ‘at 60 min’ have been added at the end of page 6.

1.3. Figure 4 to be discussed and correlated properly. Cite and compare with literature reports.

Response: Figure 4 illustrates changes in diffraction intensity during digestion of praziquantel in milk and infant formula. No such studies exist at all in the literature, even using other analytical approaches like HPLC. We do however cite a number of examples of using the approach for other drugs (refs 9, 10, 19, 22, 23, 24). We believe this to be sufficient citation of relevant studies in the absence of direct comparator studies, and discuss the results in the context of results for these other similar lipophilic drugs.

1.4.  The outcome of the study to be corelated with therapeutic outcome.

Response: The enhanced bioavailability of praziquantel with food is already well recognised in the literature as discussed in the manuscript (ref 4,17,28 cited in text). The therapeutic effect is linked to exposure, so in the context of this study, aiming to determine the potential of milk and infant formula as lipid sources to dissolve drug during digestion, a therapeutic study is not warranted.

Reviewer 2 Report (New Reviewer)

In this study, the authors revisited the dissolution of praziquantel in milk and biologically relevant media. The context of digestion was introduced to better understand the drug dissolution of praziquantel under gastrointestinal conditions. This study showed that praziquantel exhibited higher solubility in the digestion products of milk and infant formula, and that administration with praziquantel may improve oral bioavailability. I consider this study to have certain practical significance. However, I have several issues to be addressed before publishing the manuscript.

  1. In the abstract, the authors stated that praziquantel administered with milk might improve oral bioavailability, but in vitro data alone were not sufficient to support this conclusion; please add in vivo bioavailability experiments.
  2. The authors mentioned in the introduction that oral administration of praziquantel in children usually requires crushing the tablets and administering the powder as a suspension with various liquids, thus supplementing the drug group in the form of suspensions in dissolution trials was considered essential.
  3. The mechanism by which the digestive products of milk and infant formula enhance the solubility of praziquantel remains to be demonstrated.
  4. In table 1, the results of the solubility of praziquantel in different dissolution media showed a large deviation. Recommend increasing the number of test repetitions.

Author Response

2.1 In the abstract, the authors stated that praziquantel administered with milk might improve oral bioavailability, but in vitro data alone were not sufficient to support this conclusion; please add in vivo bioavailability experiments.

Response: As mentioned above and in earlier reviews, the positive food effect for praziquantel is well known. Increases in solubilisation of the drug immediately prior to absorption (such as after digestion of milk, Figures 3a and 4b) will improve absorption and consequently bioavailability. The paper sets out to explore the potential of milk and infant formula to provide that solubilisation that would, in vivo be already known to lead to improved absorption. Thus it is not appropriate to run a pre-clinical pharmacokinetic study in animals when there is nothing further to learn. A pharmacokinetic study in humans may be run in the future by the Bill and Melinda Gates Foundation who funded this work as a result of these in vitro findings.

2.2 The authors mentioned in the introduction that oral administration of praziquantel in children usually requires crushing the tablets and administering the powder as a suspension with various liquids, thus supplementing the drug group in the form of suspensions in dissolution trials was considered essential.

Response: Of course crushing the tablet will change the kinetics of dissolution (compared to e.g. Figure 2 where whole Biltricide tablets were used) however it would not change the extent of solubilisation, resulting in incomplete dissolution under some conditions and therefore not impact on the conclusions. The crushing of the tablets is done to divide the dose for young patients and to form a liquid dose form, however the Biltricide formulation also has not been approved for use in young children. The potential for milk and infant formula to provide an alternative pediatric friendly formulation was the focus here.

2.3 The mechanism by which the digestive products of milk and infant formula enhance the solubility of praziquantel remains to be demonstrated.

Response: The proposed mechanism has been clarified at the top of page 7. “Thus the mechanism proposed here is that production of fatty acids and monoglycerides as a result of lipid digestion provides a medium with much greater solubilisation capacity (dispersion + digestion profile in Figure 2a), driving dissolution of drug that otherwise would remain undissolved (e.g. dispersion only data in Figure 2a).”

2.4 In table 1, the results of the solubility of praziquantel in different dissolution media showed a large deviation. Recommend increasing the number of test repetitions.

Response: We agree that more replicates may improve the reported precision of the measurements of the data in Table 1 purely through a statistical effect (on standard deviation), however the measurement of drug solubility in milk-like systems is notoriously difficult due to the need to separate the various lipid, aqueous and pellet layers effectively (hence the more recent efforts to develop scattering methods that do not require separation). The conclusions are however very clear that digestion boosts the solubility of drug, meaning the lipids and colloidal structures formed during digestion are a more favourable environment for drug to be solubilised into. We don’t believe the addition of further replicates will lead to different conclusions and therefore do not agree that further dissolution studies are necessary.

Reviewer 3 Report (New Reviewer)

Oral route is preferred route of administration in pediatrics due to ease of administration and dose control. In recent times, milk has gained importance as a drug delivery system owing to its ability to dissolve poorly water soluble APIs. In this article, authors have tested praziquantel in variety of bio relevant media while addressing the importance of digestion as missed in previously. However, need to address following comments for better understanding.

·        In introduction, please also elaborate why commercial pediatric oral suspensions are not available.

·        Mention the previous studies related to praziquantel testing in bio relevant media.

·        Authors did not discussed the variation in drug solubilized in fasting and feed state as mention in Figure 2b.

·        Why there is variation of percentage drug solubilized in fasting and feed state as can be seen in Figure 2a and 2b. Moreover, it is contradictory to what mentioned in line 219-221. Explain

·        In table 1 solubility of drug is higher in feed state (581 ± 20  μg/mL) as compared to fasting state  (289 ± 77  μg/mL)  as expected. Nevertheless, this is not reflected in figure 1 during dissolution. Explain

·        Line 261-265: These lines could be improved to explain the effect of infant milk formulation on drug solubilization.

·        Line 308-312: Please rewrite the statement.

Author Response

 3.1 In introduction, please also elaborate why commercial pediatric oral suspensions are not available.

Response: Biltricide has not been approved for children under the age of four. It is not clear whether the market for older children would be sufficient to warrant development, or whether development never occurred for other reasons. These points have been added to the introduction on page 2. It should be noted that the approach here with infant formula would be a dry powder blend with drug for administration at the time of reconstitution so limited stability requirements would be required.

3.2 Mention the previous studies related to praziquantel testing in bio relevant media.

Response: We have cited the relevant literature in the context of incomplete dissolution in biorelevant media (reference 8 in the second paragraph), which was the issue that stimulated the project in the first instance. It is also discussed in the Results and Discussion section “Effects of co-administering praziquantel with milk on drug dissolution are however not understood and not widely explored [8,21]. Dinora et al. showed slow dissolution of praziquantel in 2.8% fat milk where only 16% drug (from 600 mg dose) was dissolved in 500/900 mL milk after 1 h, compared to about 27% and 55% in FaSSIF and FeSSIF respectively [8].”

3.3  Authors did not discussed the variation in drug solubilized in fasting and feed state as mention in Figure 2b.

Response: The differences between fed and fasted state behaviour in Figure 2b are discussed in section 3.1, both in the part starting “In addition, as depicted in Figure 2b, once praziquantel was dissolved in the low pH gastric condition, no drug precipitation occurred when the pH of the medium was adjusted to 6.8, suggesting that praziquantel could remain solubilized during gastrointestinal transit.” And the paragraph that follows that section.

3.4 Why there is variation of percentage drug solubilized in fasting and feed state as can be seen in Figure 2a and 2b. Moreover, it is contradictory to what mentioned in line 219-221. Explain

Response: We presume the reviewer is asking why fasted and fed state gives reduced dissolution compared to the USP gastric media containing SLS yet the drug shows a positive food effect. As explained in the text, the SLS provides an artificially high solubilising environment, that is not reflective of the biorelevant condition. The ‘switch’ from gastric media to Fassif in Figure 2b does not result in precipitation (as may have been anticipated) because the SLS is still present after changing pH and adding bile salts. However, it is quite clear that neither changing pH nor adding bile salt can stimulate dissolution beyond approximately 50%. This is not counter to the finding that the drug shows a positive food effect because in the case of taking a meal the digested lipid provides additional solubilisation capacity over the bile salt micelles. This point is made with additional text at the start of Section 3.2 around lines 219-221.

3.5 In table 1 solubility of drug is higher in feed state (581 ± 20  μg/mL) as compared to fasting state  (289 ± 77  μg/mL)  as expected. Nevertheless, this is not reflected in figure 1 during dissolution. Explain

Response: We assume the reviewer is referring to Figure 2a – in which case the extent of dissolution in Fessif was approximately 20% higher than in Fassif, consistent with the solubility data.

3.6 Line 261-265: These lines could be improved to explain the effect of infant milk formulation on drug solubilization.

Response: The section was extensively reworded to make the text clearer, thanks for the suggestion.

3.7 Line 308-312: Please rewrite the statement.

Response: There were residual corrections in the text from the previous review which has been removed and one phrase reworded that was not quite clear.

Reviewer 4 Report (New Reviewer)

The manuscript could be significantly improved - the referee has not been the reviewer of the first submission - but the current version still contains many more or less important points to be clarified.

The paper needs a repeated, major revision.

- Firstly, the Keywords section contains 5 of 6 items from the title! Authors who do such an execrable thing should carefully examine the content of their writings to see if they have committed a similar assassination elsewhere.

- Authors have considered emphasizing that they used a USP medium in their experiments. No reason is to write it in the Abstract. There is also no reason to write it twice, in lines 56-57 and 182-183. Neither the authors nor their work has a minimal connection to the US. Why did they select this description? Is it missing from both Eur. and Australian pharmacopeias?

- Content of lines 43-44 is repeated later (lines 313-314) using different references. Is there a reason for the repetition and the new references there?

- In line 48, "Although food (high fat/high carbohydrates) has been ...". The meaning of high fat/high carbohydrates is unclear. Is the content high? In the case of carbohydrates, the carbohydrate content is high, or the carbohydrates themselves are high molecular weight carbohydrates?

- In line 84, FaSSIF/FeSSIF/FaSSGF expressions are unresolved. Please define abbreviations upon their first appearances.

- In lines 151-152, simplify this sentence because the information is obvious. Alternatively, they could write that HPLC grade solvent is in the HPLC liquid phase since this statement is as meaningless as what they wrote.

- The expression "... specific ratios of equivalent praziquantel API to fat;..." is unclear. What fat are the authors talking about? Is that ratio a measured or calculated value?

- In lines 197-197, "... thus the solubility in gastric (0.1 M HCl) and small intestinal (50 mM phosphate buffer, pH 6.8 with no bile 196 salts) conditions were not statistically different (297±6 µg/mL and 282±149 µg/mL ..."

The referee understands that some experiments may have a high experimental error. However, 282±149 µg/ml is meaningless in solubilities (neither are 1782±1131, 1744±781, 52.4±33.3, 51.3±23.0 in Table 1). The reviewer doubts that the solubilities were not statistically different but rather that they could not confirm the different solubilities.

If the experimental error is so high (>50% of the measured value!) in a simple solubility experiment, then something is wrong with the experimenting person. A similar problem exists in Table 1 for digested milk and Digested infant formula.

- As far as the referee understood, the authors removed the insoluble parts of the tablets after disintegrating them. Did they measure the residual PQ content of those solids? If not, why?

- In lines 200-201, the authors discuss the reason for differences between the literature and their solubilities. It is a little funny. Who prevented the authors from (also) testing the solubility of PQ under the same conditions? The referee is sure that if the experimental error is ±149 µg/mL in phosphate buffer, the literature value of 206 µg/mL fits into their limits. The reviewer is also concerned that these three values will require further experiments to reduce the error to a reasonable level.

Additionally, the authors wrote the literature data but missed inserting a reference.

- The first paragraph of Section 3.2, lines 228-235. According to the manufacturer of Biltricide, the bioavailability (in adults?) is around 80%. The reviewer may reasonably assume that the manufacturer used ingested tablets rather than dissolved or suspended PQ in the dissolution/ bioavailability studies. Why do the authors conclude that Section 3.1 confirms the improved bioavailability only in young infants?

By the way, "young infant" is an underdetermined expression.

The food-assisted increased bioavailability is known, as the authors referenced it - but did not mention the studied foods. Why do the readers have to find the cited articles? The referee assumes that eliminating the useless repetitions produces enough space for inserting the foods.

- Lines 251-254 and Figure 3 raise questions.

Are the authors sure that digestion&suspension improve the PQ solubility? Can it be possible that their HPLC method was inappropriate for some peak separations?

Did the authors check the PQ chemical stability in the presence of PL or the PL solubilization power on PQ API?

Did the bitter taste disappear in the milk?

- The referee assumes that the insertion into lines 313-316 was by a previous reviewer's suggestion. The text became a long and confusing message. Some simplification of 313-316 lines would be reasonable.

- The Reference section contains some inconsistencies.

Compare refs. 9, 12, 20, 21, 25 with the others, like 4.

References 2, 3, and 18 need links.

Links 5 and 29 have doi (check out https://www.ingentaconnect.com/content/govi/pharmaz/2009/00000064/00000002/art00005 and https://www.sciencedirect.com/science/article/pii/S0169409X22000291).

Author Response

We thank reviewer 4 for their suggestions (none of which impact the Conclusions) but cannot go without comment that the reviewer may wish to consider the tone in which they provide reviews in future to reflect an appropriate level of respect and professionalism.

4.1 Firstly, the Keywords section contains 5 of 6 items from the title! Authors who do such an execrable thing should carefully examine the content of their writings to see if they have committed a similar assassination elsewhere.

Response: The keywords appropriately capture the main points of the manuscript – if they were not reflected in the title either they would be random or the title would be random, neither of which assists researchers in finding the relevant work.

4.2 Authors have considered emphasizing that they used a USP medium in their experiments. No reason is to write it in the Abstract. There is also no reason to write it twice, in lines 56-57 and 182-183. Neither the authors nor their work has a minimal connection to the US. Why did they select this description? Is it missing from both Eur. and Australian pharmacopeias?

Response: The reference was not intended to be parochial. We have changed reference to USP to ‘pharmacopoeial’.

4.3 Content of lines 43-44 is repeated later (lines 313-314) using different references. Is there a reason for the repetition and the new references there?

Response: Reference 3 is general for definition of a BCS Class II drug, not specific to praziquantel. Reference 4, which was cited as both sections, is the main reference for praziquantel itself having a positive food effect (relevant to both sections). Reference 17 and 28 are about the clinical/pharmacokinetic aspects of praziquantel (added as additional support in discussion for reference 4 in response to previous review. Reference 4 is necessary in both sections; the first section in the intro only needs one ref to make the point, while the other two are positioned in the discussion section providing additional support of the overall concept of making sure praziquantel is administered with ‘food’ in the current case milk or infant formula of sufficient fat content. The referencing therefore is appropriate in our view. Note that none of these references are self-citations.

4.4 In line 48, "Although food (high fat/high carbohydrates) has been ...". The meaning of high fat/high carbohydrates is unclear. Is the content high? In the case of carbohydrates, the carbohydrate content is high, or the carbohydrates themselves are high molecular weight carbohydrates?

Response: In hindsight there was no need to elaborate to make the point, so the sentence has been modified to read “Although food has been shown to improve the oral bioavailability of praziquantel in adult volunteers, translating the findings to infants can be challenging given the different gastrointestinal physiology.”

4.5 In line 84, FaSSIF/FeSSIF/FaSSGF expressions are unresolved. Please define abbreviations upon their first appearances.

Response: The abbreviations have been defined at that point in the manuscript.

4.6 In lines 151-152, simplify this sentence because the information is obvious. Alternatively, they could write that HPLC grade solvent is in the HPLC liquid phase since this statement is as meaningless as what they wrote.

Response: We cannot identify the sentence being referred to – line 151-152 has the sentence: “Praziquantel standards were prepared using praziquantel API, where a stock solution (1 mg/mL in 1-methyl-2-pyrrolidinone) was diluted with the mobile phase to a final concentration range of 0.1-40 µg/mL.” This sentence seems fine to us – it may be excessively clear to say that praziquantel API was used to prepare the standards but that is to clearly differentiate from using e.g. Biltricide to make standards which many groups do obviously ignoring issues of overage etc. Perhaps a directly formed question would have made it clearer what was being commented on.

4.7 The expression "... specific ratios of equivalent praziquantel API to fat;..." is unclear. What fat are the authors talking about? Is that ratio a measured or calculated value?

Response: The known fat content of milk and infant formula allows the ratio of drug to total fat in the liquid to be controlled, to understand the effect of fat content on drug solubilisation during digestion (see Figure 4b). We have slightly reworded the sentence to make this clearer.

4.8 In lines 197-197, "... thus the solubility in gastric (0.1 M HCl) and small intestinal (50 mM phosphate buffer, pH 6.8 with no bile 196 salts) conditions were not statistically different (297±6 µg/mL and 282±149 µg/mL ..." The referee understands that some experiments may have a high experimental error. However, 282±149 µg/ml is meaningless in solubilities (neither are 1782±1131, 1744±781, 52.4±33.3, 51.3±23.0 in Table 1). The reviewer doubts that the solubilities were not statistically different but rather that they could not confirm the different solubilities. If the experimental error is so high (>50% of the measured value!) in a simple solubility experiment, then something is wrong with the experimenting person. A similar problem exists in Table 1 for digested milk and Digested infant formula.

Response - As is always the case, more precision can be achieved in any experiment given sufficient time and artificially by more replicates. Whether the effort is rewarded by changing the conclusion is a different matter. Measuring solubility analytically in digested milk and infant formula is a technically very challenging task (not a simple solubility experiment) - due to the phase separations that occur when trying to pellet the undissolved drug and is also complicated by calcium soap formation – this was the initial motivation behind development of the complimentary SAXS methodology as no separation is necessary to detect undissolved drug. The lipid and aqueous phases that separately constitute the supernatant are separately assayed and the error associated with each propagates as they are added together to obtain an overall solubility value. Thus, the values overall are way higher in digested media but with high error associated with the measurement – this supports at least the expectation that drug will be better solubilised in during digestion in milk and infant formula than in undigested media. The authors accept the reviewers point about the error but it does not change the conclusion and the additional effort required to obtain greater precision will not change the outcome.

4.9 As far as the referee understood, the authors removed the insoluble parts of the tablets after disintegrating them. Did they measure the residual PQ content of those solids? If not, why?

Response: We presume the reviewer refers to the filtration of samples taken during dissolution studies – this was of course to remove undissolved drug (as well as any insoluble excipients) from the dissolved drug as is common practice in dissolution studies.

4.10 In lines 200-201, the authors discuss the reason for differences between the literature and their solubilities. It is a little funny. Who prevented the authors from (also) testing the solubility of PQ under the same conditions? The referee is sure that if the experimental error is ±149 µg/mL in phosphate buffer, the literature value of 206 µg/mL fits into their limits. The reviewer is also concerned that these three values will require further experiments to reduce the error to a reasonable level.

Response – The sentence has been modified to now read “Our findings appear slightly higher but probably not significantly different to those reported in the literature where the solubility of praziquantel…”. We had good (better) reasons for selecting the media in which we studied the dissolution and solubility, the comparison to the literature values is not intended to verify or otherwise the accuracy either way, and as the reviewer points out they are probably indistinguishable. Again the conclusion that the solubilities are low compared to pre-digested milk or infant formula does not change regardless of the precision of the studies here compared to others.

4.11 Additionally, the authors wrote the literature data but missed inserting a reference.

Response - Reference 14 at the end of the sentence refers to the literature data – the citation has been moved to earlier in the sentence.

4.12 The first paragraph of Section 3.2, lines 228-235. According to the manufacturer of Biltricide, the bioavailability (in adults?) is around 80%. The reviewer may reasonably assume that the manufacturer used ingested tablets rather than dissolved or suspended PQ in the dissolution/ bioavailability studies. Why do the authors conclude that Section 3.1 confirms the improved bioavailability only in young infants?

Response: The manuscript is not claiming that the bioavailability of praziquantel in the form of Biltricide (not approved in infants) is low. It is illustrating that if dissolution is the indicator of likely improved bioavailability then milk and infant formula may have a role to play in reformulation of praziquantel for infants.

4.13 By the way, "young infant" is an underdetermined expression.

Response: The word young has been removed as suggested.

4.14 The food-assisted increased bioavailability is known, as the authors referenced it - but did not mention the studied foods. Why do the readers have to find the cited articles? The referee assumes that eliminating the useless repetitions produces enough space for inserting the foods.

Response - It is not necessary to cite the full details of the food in a research article (as opposed to a review of using food to increase bioavailability). For example, reference 7 states the full description of the meal as “The high-fat diet consisted of two fried eggs, one slice of ham, orange juice, and milk (200 ml) (protein, 19.63%; fat, 32.44%; and carbohydrate, 47.91%; 656 cal); the high-carbohydrate diet consisted of four tortillas, tomato, chicken (100 g), a slice of white bread, and a glass (200 ml) of orange juice (protein, 15.30%; fat, 10.54%, and carbohydrate, 74.15%; 674.5 cal).” Considering it was not work done in this study, it is very unlikely that the reader wants that level of detail. As a compromise, we have modified reference to food where it is specific to a particular study to say “high fat or high carbohydrate meal”.

4.15 Lines 251-254 and Figure 3 raise questions. Are the authors sure that digestion&suspension improve the PQ solubility? Can it be possible that their HPLC method was inappropriate for some peak separations?

Response – It is not clear what the reviewer is asking here – the data in Figure 3a clearly illustrate the dependence of digestion on the solubilisation of drug dynamically compared to just dispersion. The dispersion data agree broadly with the solubility data that drug is soluble to a degree in undigested  milk and more so in undigested infant formula but that digestion boosts dissolution by providing additional capacity, the difference being greater for milk which also is consistent. So the answers to the questions as written is Yes and No.

4.17 Did the authors check the PQ chemical stability in the presence of PL or the PL solubilization power on PQ API?

Response: Stability of praziquantel in bile salt/phospholipid mixtures (assuming PL is referred to phospholipid) was not separately assessed. However, the fact that the digestion studies indicated ~100% dissolution provided confidence that minimal if any degradation of praziquantel had occurred during the digestion process.

4.18 Did the bitter taste disappear in the milk?

Response: This is an interesting question, clearly a clinical taste trial is not in the remit of this physical-chemical based study but would be an obvious hurdle for any oral liquid preparation.

4.19 The referee assumes that the insertion into lines 313-316 was by a previous reviewer's suggestion. The text became a long and confusing message. Some simplification of 313-316 lines would be reasonable.

Response: The sentence has been split into two to enhance clarity.

4.20 The Reference section contains some inconsistencies.

Compare refs. 9, 12, 20, 21, 25 with the others, like 4.

References 2, 3, and 18 need links.

Links 5 and 29 have doi (check out https://www.ingentaconnect.com/content/govi/pharmaz/2009/00000064/00000002/art00005 and https://www.sciencedirect.com/science/article/pii/S0169409X22000291).

Response: It was not immediately clear to the authors what the referee meant by compare refs. Links to references 2, 3, 18 and the DOI for references 5 and 29 have been added to the manuscript.

Round 2

Reviewer 1 Report (Previous Reviewer 5)

Figure 2 and 3, In vitro dissolution studies on pharmaceutical formulations is an indicative of cumulative percent drug release from the formulation and the graph should also indicate the same.

Author Response

Figure 2 and 3, In vitro dissolution studies on pharmaceutical formulations is an indicative of cumulative percent drug release from the formulation and the graph should also indicate the same.

Response: Y-axis for both figure 2 and 3 have been changed to “cumulative % drug solubilised”

Reviewer 2 Report (New Reviewer)

The authors have provided reasonable responses to all issues. After correcting the format of the references, I consider the manuscript suitable for publication.

Author Response

The authors have provided reasonable responses to all issues. After correcting the format of the references, I consider the manuscript suitable for publication.

Response: The reference list has been double checked, and will be also confirmed during publication phase.

Reviewer 4 Report (New Reviewer)

The authors corrected almost all issues recommended by the referees.

Unfortunately, many inconsistencies in the reference section are still uncorrected!

The referee would suggest finding and using correct abbreviations for all the journal names. DOI in refs. 9, 12, 20, 21, and 25 are in different formats than the others. In ref. 29, the doi character size differs from the others.

Authors also have to decide between "sulf..." and "sulph...*".

The manuscript, therefore, needs a minor revision but is otherwise suitable for publication.

Author Response

4.1 The referee would suggest finding and using correct abbreviations for all the journal names. DOI in refs. 9, 12, 20, 21, and 25 are in different formats than the others. In ref. 29, the doi character size differs from the others.

Response: DOI formats are now consistent and the journal names have been updated.

4.2 Authors also have to decide between "sulf..." and "sulph...*".

Response: Sulphate has been changed to sulfate

This manuscript is a resubmission of an earlier submission. The following is a list of the peer review reports and author responses from that submission.

Round 1

Reviewer 1 Report

Milk as a carrier of poorly soluble drugs is meaningful for the development of pediatric drugs. This article investigated the dissolution of praziquantel in biorelevant media and milk, which is helpful for milk-based oral pediatric formulations.

There have been many reports on oral digestion and absorption studies of lipid formulations, especially the development of in vitro digestion, dissolution or absorption models of lipid formulations. It would be better if the authors could incorporate previous findings on in vitro digestion and dissolution of lipid formulations when discussing data from in vitro digestion experiments.

Author Response

It would be better if the authors could incorporate previous findings on in vitro digestion and dissolution of lipid formulations when discussing data from in vitro digestion experiments.

This is, to our knowledge, the first study that incorporates an in vitro digestion step to characterise solubilisation of praziquantel in milk-based lipid formulations. We have reported and cited the studies of a similar nature with other drugs but don’t believe that it is appropriate to more widely review all papers in the field of lipid based formulations.

Reviewer 2 Report

The overall manuscript flows well, with good scientific evidences.

How relevant is the study considering the dosage form is a significantly large tablet containing 600 mg API intended for infants. 

How was the concentration of Pancreatic lipase decided for the study, and what would be the ideal concentration difference between adults and infants

What was the source of the baby milk? What were the excipients in the infant formula?

The Materials in section 2.1 could be put into a Table form, might be easier to visualize

Line:187: The term "overestimated" seems unclear what it means

Author Response

2.1 The overall manuscript flows well, with good scientific evidences. How relevant is the study considering the dosage form is a significantly large tablet containing 600 mg API intended for infants.

We thank the reviewer for the comments. The large tablet containing 600 mg API is not intended for infants, neither is the volume of the USP dissolution apparatus. However, in this study, we aim to determine the amount of drug that can be solubilised in a given volume of milk and digested milk (of known fat content) and as such, is not dependent on the adult vs infant dosage and would be scaled accordingly. Note that there is no current suitable paediatric-specific dose form for praziquantel, so this represents the first step towards understanding the impact of digestion of milk-like formulations on drug dissolution.

2.2 How was the concentration of Pancreatic lipase decided for the study, and what would be the ideal concentration difference between adults and infants

The amount of pancreatic lipase added was designed to be in excess to circumvent enzyme-limited lipid digestion hence drug solubilisation. Activity of pancreatic lipase in adults and infants are highly variable in nature. In adults for example, about 84-1224 tributyrin unit/mL lipase activity has been reported in the fasted state whereas about 149-586 tributyrin unit/mL was observed in infants. These were subsequently altered in the fed-state where the lipase activity depended on the time after feeding. (Pediat. Res. 1978, 12: 631-634) An ideal concentration difference between adults and infants is therefore difficult to ascertain but comparison between in vitro digestion in adults vs infants can potentially be made with harmonisation of the digestion protocols such as that developed by INFOGEST.

2.3 What was the source of the baby milk? What were the excipients in the infant formula?

Nutritional information for milk and infant formula has been added to Table S1 in the supporting information section. The brand of infant formula cannot be disclosed for commercial reasons, and we have published several papers previously providing this level of nutritional information.

Table S1. Nutritional information of milk and infant formula (IF; based on reconstitution of 14.0 g powder) per 100 mL.

Nutritional information

Quantity per 100 mL

Milk

IF

Total fat

3.8 g

3.8 g

Saturated fat

2.5 g

2.1 g

Total protein

3.4 g

1.5 g

Carbohydrate

4.8 g

7.9 g

Sodium

40 mg

36 mg

Calcium

115 mg

60 mg

Vitamin A

41 µg

60 µg

Riboflavin (vitamin B2)

200 µg

112 µg

2.4 The Materials in section 2.1 could be put into a Table form, might be easier to visualize

We thank the reviewer for the comment. However, to align with the format typical in MDPI Pharmaceutics published articles, we have decided to keep the materials in material section as is and no changes are made to the text.

2.5 Line:187: The term "overestimated" seems unclear what it means

We apologise for the confusion. The overestimation was pointed to the amount of praziquantel solubilized when SLS was added to the dissolution medium (0.1 M HCl) vs that in 0.1 M HCl only. The word “overestimated” was changed to “increased”.

Reviewer 3 Report

I have reviewed through the whole manuscript critically and I found that authors have attempted a good work about the potential positive impact of digestion of milk on praziquantel dissolution. However, some minor and major issues need to be addressed:

  1. Line 242: the impact of digestion “on” drug solubilization
  2. Line 273: Have authors conducted peak area analysis at other peaks, like q=1.08, 1.15, which showed similar peak intensity compared to q=1.2? The authors should show more analysis data at multiple peaks to eliminate the bias.
  3. Line 281-283: Please explain why 58.5 mg and 87.7 mg drug/g milk fat were specifically selected. For example, incomplete solubilization was observed at 90 mg drug/g milk fat but complete drug solubilization was achieved at 50 mg drug/g milk fat, then this range would be 50-90 mg/drug/g milk fat. What is the critical message behind this SAXS peak area analysis?
  4. One typo: solubilisation could be “solubilization” e.g. in line 280 and line 281
  5. Line 317-318: It’s necessary for authors to provide any in vivo data or IVIVC discussions based on lipid formula (e.g. milk vs. digestible milk) for praziquantel or other BCS II compounds to support the highlight of this paper, “Incorporating a digestion step into in vitro dissolution testing, particularly for digestible lipid-based formulations is therefore likely to be critical to establish a reflective link between in vitro dissolution/digestion and in vivo exposure.”

Author Response

3.1 Line 242: the impact of digestion “on” drug solubilization

Changes has been made accordingly.

3.2 Line 273: Have authors conducted peak area analysis at other peaks, like q=1.08, 1.15, which showed similar peak intensity compared to q=1.2? The authors should show more analysis data at multiple peaks to eliminate the bias.

Peak area analysis of the drug at different peak positions show similar trend of drug solubilisation (an example is shown in figure below). The following texts have been added to the main manuscript line 263-264 for clarification:

“Similar trend in drug solubilization was observed across different peak positions, an example of which is shown in Figure S1.”

Caption for attached figure: Figure S1. X-ray scattering pattern of praziquantel in 3.8% infant formula during dissolution (time < 0 min) and digestion (time > 0 min) showing the progressive decrease in peak intensity that is synchronous across peaks at different positions.

3.3 Line 281-283: Please explain why 58.5 mg and 87.7 mg drug/g milk fat were specifically selected. For example, incomplete solubilization was observed at 90 mg drug/g milk fat but complete drug solubilization was achieved at 50 mg drug/g milk fat, then this range would be 50-90 mg/drug/g milk fat. What is the critical message behind this SAXS peak area analysis?

The aim of tailoring the different drug-to-fat ratio was to assess how much drug can be solubilised in a given amount of fat. This information is critical to the design of milk-based lipid formulation such that an appropriate amount of fat (in milk) can be co-administered with a specific drug dose to achieve a desired level of drug solubilisation. Therefore, it does not matter specifically which mg/g fat was tested but rather the range onto which the drug can be soluble. Due to the limited beamtime awarded by the Australian synchrotron, only a finite number of ratios were able to be investigated to evaluate the range onto which a maximum amount of drug can be soluble.

3.4 One typo: solubilisation could be “solubilization” e.g. in line 280 and line 281

“Solubilisation” has been changed to “Solubilization” throughout the manuscript

3.5 Line 317-318: It’s necessary for authors to provide any in vivo data or IVIVC discussions based on lipid formula (e.g. milk vs. digestible milk) for praziquantel or other BCS II compounds to support the highlight of this paper, “Incorporating a digestion step into in vitro dissolution testing, particularly for digestible lipid-based formulations is therefore likely to be critical to establish a reflective link between in vitro dissolution/digestion and in vivo exposure.”

The authors agree that without IVIVC, a conclusive evidence that an increased oral bioavailability of praziquantel can be achieved when co-administered with milk or infant formula cannot be drawn. This work will constitute our future studies. Enhancement of in vitro drug solubilisation during digestion and the correlation with an increased in vivo oral bioavailability has however been observed in our previous studies on other poorly water-soluble drugs including artefenomel and ferroquine. Nevertheless, the following text has been added to the manuscript (conclusions section, line 307-308) to clarify:

“Incorporating a digestion step into in vitro dissolution testing, particularly for digestible lipid-based formulations is therefore likely to be critical to establish a reflective link between in vitro dissolution/digestion and in vivo exposure but further in vitro in vivo correlation (IVIVC) studies are needed to confirm the relationship.”

Reviewer 4 Report

The manuscript by Ramirez et al. revisits the dissolution of praziquantel in biorelevant media and milk, as well as the impact of milk digestion on drug dissolution to better understand drug solubilization and dissolution behaviors in vivo. It was found that while praziquantel could be solubilized to a limited extent into bile salt/phospholipid micelles, the drug exhibited higher solubilization capacity into the digestion products of milk and infant formula. Therefore, incorporating a digestion step into in vitro dissolution testing is critical to establish a reflective link between in vitro dissolution/digestion and in vivo exposure.

The manuscript is well written and would be of interest to the readers of Pharmaceutics. 

Please find below a couple of comments and suggestions to improve the quality of the manuscript.

1. L83-36: The compositions of the infant formula should be provided in the Supplementary Material of the manuscript.

2. L100 and L116: The authors should explain in the manuscript the discrepancy in the paddle speeds (50 rpm vs. 200 rpm).

3. Figure 2a 

- Please discuss why the maximal praziquantel concentration in FeSSIF (50%*600 mg/900 mL ≈ 0.333 mg/mL = 333 µg/mL) was much lower than the solubility of the drug in this biorelevant medium (581 ± 20 µg/mL, Table 1)?

- Why the release of praziquantel in phosphate pH 6.8 was lower than in 0.1 M HCl despite the fact that the solubility of the drug in these two media were more or less the same (Table 1)?

- L209-211: please discuss why “a greater amount of praziquantel was solubilized in FeSSIF compared to FaSSIF and phosphate buffer; whilst no significant differences were observed between FaSSIF and phosphate buffer”? Similarly, please discuss in the manuscript why the solubility of praziquantel in the phosphate buffer was much lower than in FeSSIF but was comparable to that in FaSSIF (Table 1)?

4. Figure 2b

- The pH switching dissolution method should be clearly described in the Materials and Methods section.

- Why the maximal praziquantel concentration in FeSSIF (60%*600 mg/900 mL = 0.4 mg/mL = 400 µg/mL) was much lower than the solubility of the drug in this biorelevant medium (581 ± 20 µg/mL, Table 1)?

5. Figure 3b: Please discuss why the % drug solubilized in the “dispersion only” profile suddenly increased between 90 min to 120 min?

6. Figure 6b: Error bars should be provided. 

7. L52-54: Please discuss the discrepancy in the results presented in the manuscript vs. the findings in the reference #8.

Author Response

4.1 L83-36: The compositions of the infant formula should be provided in the Supplementary Material of the manuscript.

Nutritional information for milk and infant formula has been added to Table S1 in the supporting information section.

Table S1. Nutritional information of milk and infant formula (IF; based on reconstitution of 14.0 g powder) per 100 mL.

Nutritional information

Quantity per 100 mL

Milk

IF

Total fat

3.8 g

3.8 g

Saturated fat

2.5 g

2.1 g

Total protein

3.4 g

1.5 g

Carbohydrate

4.8 g

7.9 g

Sodium

40 mg

36 mg

Calcium

115 mg

60 mg

Vitamin A

41 µg

60 µg

Riboflavin (vitamin B2)

200 µg

112 µg

4.2 L100 and L116: The authors should explain in the manuscript the discrepancy in the paddle speeds (50 rpm vs. 200 rpm).

Additional text has been added, which now reads:

“Paddle speed was set to 200 rpm to enhance mixing of the milk dispersion, which has a higher viscosity relative to buffer.”

4.3 Figure 2a

4.3.1 Please discuss why the maximal praziquantel concentration in FeSSIF (50%*600 mg/900 mL ≈ 0.333 mg/mL = 333 µg/mL) was much lower than the solubility of the drug in this biorelevant medium (581 ± 20 µg/mL, Table 1)?

4.3.2 Why the release of praziquantel in phosphate pH 6.8 was lower than in 0.1 M HCl despite the fact that the solubility of the drug in these two media were more or less the same (Table 1)?

The release was slightly lower in phosphate buffer because the solubility was lower – there was a larger error associated with the measurement of solubility in phosphate buffer we believe due to wetting and although the average solubility was slightly lower, more extensive determination likely would yield a lower average we believe from experience with similar compounds.

4.3.3 L209-211: please discuss why “a greater amount of praziquantel was solubilized in FeSSIF compared to FaSSIF and phosphate buffer; whilst no significant differences were observed between FaSSIF and phosphate buffer”? Similarly, please discuss in the manuscript why the solubility of praziquantel in the phosphate buffer was much lower than in FeSSIF but was comparable to that in FaSSIF (Table 1)?

Additional text has been added, which now reads:

“These findings suggest that secretion of bile salts at fed state concentrations may increase the apparent solubility of praziquantel and that micelles in FeSSIF [15] can provide a greater capacity for drug to be solubilized compared to vesicles in FaSSIF [15].”

4.4 Figure 2b

4.4.1 The pH switching dissolution method should be clearly described in the Materials and Methods section.

The following text has been added to manuscript section 2.2:

Dissolution of Biltricide® in the gastric medium followed by pH switch to small intestinal pH at 6.8 was also performed. Biltricide® tablet (1 tablet) was dispersed in the gastric medium (either 800 mL of 0.1 M HCl in water with 2 mg/mL SLS or 900 mL of FeSSIF pH 1.2) for 1 hour after which the pH was adjusted to 6.8 using 5 M NaOH. Following pH adjustment, 80 mL of 10 × concentrated  FaSSIF was added to the former (0.1 M HCl with 2 mg/mL SLS). Samples were collected and filtered as described above prior to further analysis.

4.4.2 Why the maximal praziquantel concentration in FeSSIF (60%*600 mg/900 mL = 0.4 mg/mL = 400 µg/mL) was much lower than the solubility of the drug in this biorelevant medium (581 ± 20 µg/mL, Table 1)?

Taking first the response to basically the same question in section 4.3 as applying here, additionally the amount of drug solubilised in FeSSIF Figure 2b was based on pH 1.2 while that in the equilibrium solubility study was based on pH 6.8.

4.5 Figure 3b: Please discuss why the % drug solubilized in the “dispersion only” profile suddenly increased between 90 min to 120 min?

The % drug solubilized at 90 and 120 minutes was not statistically significant. Student t-test reveals a p-value of 0.08.

4.6 Figure 6b: Error bars should be provided.

There are four figures in the manuscript and we believe the reviewer is pointing to Figure 4b. Experiments performed at the synchrotron-SAXS were not replicates due to the limited beamtime awarded hence no error bars were included. Previous experience has shown this process to be highly reproducible and also agree with orthogonal techniques such as HPLC or low frequency Raman scattering, but it is not possible to run replicate data due to this constraint. The following text has been added to the figure caption for clarification. “Changes in peak area during digestion for the diffraction peak at q = 1.20 Å-1 (* in panel (a)) in FaSSIF, milk and infant formula at different drug to fat ratios (n=1).”

4.7 L52-54: Please discuss the discrepancy in the results presented in the manuscript vs. the findings in the reference #8.

Reference 8 investigated the dissolution of praziquantel in 2.8% bovine milk, FaSSIF, FeSSIF and USP-medium, similar to our study. The percentage release of praziquantel at 1 hr was in the order of milk < FaSSIF < FeSSIF < USP-medium.  Despite the slight discrepancy in the dissolution behaviour of milk (where we used 3.8% instead of 2.8%), similar trend was observed where highest drug dissolution was observed in the USP-medium compared to FeSSIF and FaSSIF. It is however worth noting that the reference study did not include a digestion of the milk media, which we show to significantly influence the extent to which praziquantel can be solubilised.

Reviewer 5 Report

The conclusion reported by the researchers is given as under:

In milk, significant dissolution of praziquantel was observed 26 only during digestion and not during dispersion hence suggesting that (1) milk can be potentially administered with praziquantel to improve oral bioavailability and (2) incorporating a digestion step into existing in vitro dissolution testing can better reflect the potential for a positive food effect when lipids are present.

However without a significant IVIVC it is too early to conclude that oral bioavailability will be enhanced as both solubility and permeability are important for systemic bioavailability of drugs after oral administration.

Secondly presence of food/lipids are only exhibiting the in vitro dissolution. Again IVIVC or in silico modeling tool could be afirmative in substantiating the claim.

Author Response

5.1 The conclusion reported by the researchers is given as under: In milk, significant dissolution of praziquantel was observed only during digestion and not during dispersion hence suggesting that (1) milk can be potentially administered with praziquantel to improve oral bioavailability and (2) incorporating a digestion step into existing in vitro dissolution testing can better reflect the potential for a positive food effect when lipids are present.

However without a significant IVIVC it is too early to conclude that oral bioavailability will be enhanced as both solubility and permeability are important for systemic bioavailability of drugs after oral administration.

5.2 Secondly presence of food/lipids are only exhibiting the in vitro dissolution. Again IVIVC or in silico modeling tool could be afirmative in substantiating the claim.

The authors agree that without IVIVC, a conclusive evidence that an increased oral bioavailability of praziquantel can be achieved when co-administered with milk or infant formula cannot be drawn. This work will constitute our future studies. Enhancement of in vitro drug solubilisation during digestion and the correlation with an increased in vivo oral bioavailability has however been observed in our previous studies on other poorly water-soluble drugs. Nevertheless, the following text has been added to the manuscript (conclusions section, line 307-308) to clarify:

“Incorporating a digestion step into in vitro dissolution testing, particularly for digestible lipid-based formulations is therefore likely to be critical to establish a reflective link between in vitro dissolution/digestion and in vivo exposure but further in vitro in vivo correlation (IVIVC) studies are needed to confirm the relationship.”

Round 2

Reviewer 3 Report

The authors have addressed my comments except the critical one: no in vivo data or IVIVC literature in regards to lipid formula of praziquantel have been discussed in this manuscript. It's common sense that monoglycerides and fatty acids produced by milk digestion can emulsify and enhance drug solubilization. However, no related in vivo evidence or even literature review has been introduced in this paper to substantiate the claims about "improve oral bioavailability" or "critical to establish a reflective IVIVC link". I have to reject it so far and hope the authors could supplement more discussions on the in vivo studies.

Reviewer 5 Report

In vitro dissolution testing has no meaning until the results are not corelated with the in vivo performance.